# The Satisfactions, Contributions, and Opportunities of Women Academics in the Framework of Sustainable Leadership: A Case Study

Gladys Merma-Molina [1] , Mayra Urrea-Solano [1,*] , Salvador Baena-Morales [2,*] and Diego Gavilán-Martín [1]

1 GIDU-EDUTIC/IN Research Group, Department of General and Specific Didactics, Faculty of Education, University of Alicante, 03690 San Vicente del Raspeig, Spain; gladys.merma@ua.es (G.M.-M.); diego.gavilan@ua.es (D.G.-M.)
2 EDUCAPHYS Research Group, Department of General and Specific Didactics, Faculty of Education, University of Alicante, 03690 San Vicente del Raspeig, Spain
* Correspondence: mayra.urrea@ua.es (M.U.-S.); salvador.baena@ua.es (S.B.-M.)

**Abstract:** Women's empowerment is one of the targets of Sustainable Development Goal 5, gender equality. However, little research has highlighted the contributions of sustainable female leadership in academic governance. In order to fill this gap, this study identifies and analyses the satisfactions, opportunities, and contributions of women academics to university governance and their perceptions of the potential impact of gender in this process. Forty-eight women leaders participated in the study. A purposive sampling technique was used because the research involved leaders who had held a management position in the university. The research methodology was qualitative, the instrument used for the collection of information was a semi-structured interview, and the analysis of the narratives was carried out with Aquad v. 7 software (Günter Huber, Tübingen, Alemania). The study revealed that the leadership style of the female academics is framed within the sustainable leadership approach. Beyond personal satisfactions, the main reward derived from the performance of the position lies in becoming transformative and catalysing agents of the institution, who try to find a balance between the economic and social interests of the organisation. The functions these female academics perform, within the framework of sustainable development, have a technical and, at the same time, humanised vision, as they focus on people and on personal and social values. Gender issues have not been a determinant in the satisfactions, opportunities, and contributions that the leaders make to the institution; however, participants emphasise that this was a strength for leadership.

**Keywords:** sustainable leadership; governance; Sustainable Development Goals; sustainable women's leadership; university; equality gender

## 1. Introduction

From a theoretical perspective, sustainable leadership (SL) is a combination of different leadership approaches in the context of sustainable development [1]. It is based on equity, and its central objectives are: (1) to achieve a balance between economic, social, and environmental interests in the organisation, and (2) to lead the institution and its members towards sustainable development by adopting and implementing socially responsible activities and strategies [2]. This basically implies that sustainable leaders have to be able to implement sustainable development policies in their organisations [3], solve problems, and drive a sustainable institution, promoting behaviours and practices that benefit all stakeholders, including future generations [4–6]. SL assumes that issues linked to sustainability are interrelated, that they cannot be addressed in a piecemeal fashion, and that any member of the organisation can take responsibility for fostering sustainable environments.

In the last two decades, there has been some interest in academia in investigating the set of competencies and values that makes a leader committed to leadership in terms of seeking to preserve the planet [7,8]. In fact, Visser and Courtice [1] have already addressed SL at the individual level through a model based on situational leadership [9,10], which is based on three dimensions: the individual as a leader (with their traits, styles, skills, and knowledge), the internal and external context of leadership, and the internal and external actions of leadership. In accordance with these dimensions, sustainable leaders should have general skills (emotional intelligence and a caring attitude, a concern for organisational culture, pursuit of goal achievement, and the ability to generate trust) and specific ones that contribute to sustainable practices (an inclusive, fair, and impartial style) [1], systemic and interdisciplinary understanding, willingness to innovate and a long-term perspective on impacts, respect for ethical principles [11], a capacity for the pursuit of goal achievement, systems thinking [7], innovative thinking [12], vision [13], creativity [14], and altruism [15]). With these skills, leaders are able to guide organisations towards a more sustainable state by facing complexity and conflicts in the social, economic, and environmental domains [16].

In addition to individual characteristics, leadership context and internal and external actions are also determining factors in SL [17]. In fact, Visser and Courtice [1] argue that the SL must take into account the external context (ecological, economic, political, cultural, and social) and the internal context (organisational culture). Added to this, the sustainable leader has to be consistent in both internal and external actions. This means that the typical internal actions of the organisation—for instance, making informed decisions, empowering people, and incorporating learning and innovation—must be aligned with sustainability challenges and opportunities through external actions, such as fostering cross-sector partnerships, or contributing to the awareness of sustainability in society.

Research on women's leadership in academia is in its infancy and has focused on highlighting the absence of women in top decision-making positions. In this vein, Abalkhail [18] conducts a study in which she examines female managers' perceptions of the factors that influence their rise to leadership positions in Saudi Arabian higher education. The author concludes that women face a number of challenges that prevent them from achieving equal representation in relevant decision-making positions. In addition, she analyses in detail legal, socio-economic, and cultural aspects that are reflected in organisational practices and that limit women's access to university leadership positions. Jane et al. [19] interviewed 35 women academics to investigate women's leadership in higher education in Hong Kong and concluded that there are institutional gaps that limit equality. The study shows the tensions women academics experience in living with and within political contexts. Moreover, Burkinshaw and White [20], through the analysis of two case studies in the context of Australian and UK Higher Education, conclude that gender power relations in universities maintain entrenched inequalities. Therefore, and taking into account the growing resistance to women's participation in leadership, especially from the younger generation, they argue that it is the universities themselves that must be corrected and not the women. In the Spanish context, Gallego-Morón and Montes López [21] analyse the influence of the organisation itself on the careers of female academics. To do so, they conducted 48 interviews with men and women, and concluded that there is a tendency towards homosociability. This means that the traditional masculinisation of power networks favours the trajectories of men and negatively affects the trajectories of women. Along the same lines, Campanini and Pizarro [22] investigate how the formal and informal norms of the university impact on gender relations, particularly for those in leadership positions. To this end, they interviewed members of the board of directors of the University of Deusto (northern Spain) and confirmed the impact caused by gender dynamics at both the individual and institutional level. While the authors recognise the importance of formal institutional commitments to gender equality (e.g., legislation), they highlight the importance of informal environments (customs, traditions, etc.), as they have a decisive influence on the implementation of formal policies.

### 1.1. Women's Sustainable Leadership

Gender-responsive sustainable development is essential in the current global context, where issues such as poverty, war, and environmental degradation are expected to persist, and there is an urgent need to do things differently [23]. Nevertheless, in the area of leadership, women still face major barriers to becoming leaders. These include unequal access to economic opportunities, lack of political representation, barriers in access to education, stereotypes that limit their performance in important positions, as well as weak self-expectation and devaluation of their successes [24–28].

Barriers to women's leadership have been addressed theoretically from both a cultural and institutional perspective [29]. According to the latter, when genders develop outside expected socio-institutional norms, women and men may engage in negative attitudes [30]. For this reason, leadership skills, such as the ability to delegate, confidence, and assertiveness, are often considered masculine qualities [31], and women are often even less liked when they are successful leaders, especially in male-dominated fields [32]. Consequently, female leaders have to walk a fine line where there are a number of traditionally masculine-prevailing traits [33]. Therefore, ascending to leadership positions remains difficult for women, and most organisations, even educational ones, still have leadership based on gender differences.

The theory of gendered organisations makes gender inequality in leadership explicit [34]. It recognises that workplaces, such as in the university context, are not gender-neutral, but are spaces where gender is deeply embedded, as power networks favour men [35]. In such contexts, women may be seen as weak and incompetent in leadership roles due to culturally defined gender norms [36]. This can cause them, among other things, to internalise doubts, mistrust themselves, and, thus, question their own leadership ability [37].

Traditional gender roles are also an additional barrier for women who have a responsibility for family and domestic affairs [38]. Women, on average, still spend proportionally more time than men in the household and in raising children [39]. Because of these imbalances, women advance more slowly in their careers and are less encouraged to take on leadership positions [40]. In short, women's paths to leadership in sustainable development are not clearly mapped out. While the literature provides guidance on what we expect to be the main barriers and contributions of women leaders in sustainable development, there is still a need for a deeper understanding of their personal perceptions and perspectives, since, as recognised by the UN General Assembly [41], their participation as leaders in sustainable development is critical.

### 1.2. Satisfactions and Opportunities in the SL

SL can be an extraordinary opportunity for the professional and personal enrichment of the leader [42]. Despite this, the opportunities and satisfactions that female sustainable leadership (FSL) generates in women have been scarcely investigated. At the professional level, some of the benefits derived from FSL, in addition to professional prestige and recognition, include greater integration into the university community, knowledge of the institution from another perspective, and the discovery of the organisation's operating mechanisms. To this must be added the high degree of satisfaction generated by having been able to contribute to the improvement and development of the institution [43].

In relation to personal satisfactions, Bianchi et al. [44] argue that women seek personal satisfaction more than monetary reward. Furthermore, Divya et al. [45] add that there are a number of emotional elements that have to do, for example, with the female leader's commitment to the organisation and to the people she leads. With FSL, the achievement of planned goals through a technical vision is humanised, and is brought closer to people, thus bringing together technical and personal dimensions in a farsighted, friendly, and lasting harmony [46]. In this vein, Acker [47] stresses that professional experience allows leaders to overcome their internal barriers and limitations, in order to discover their true leadership skills and to acquire greater self-confidence. Those who are mothers also see the

managerial role as an opportunity of extraordinary value to offer a model and a reference at a professional level for their children [48]. However, according to Armstrong [49], this responds more to an adaptation mechanism to achieve a balance between family and professional life without remorse or frustration, as it is usually one of the strategies most used by women to alleviate their feelings of guilt for violating the mandates of traditional gender roles.

### 1.3. SL Contributions to the Institution

The sustainable leader can effectively contribute to the transformation and progress of an organisation [50,51], as long as their roles are focussed on addressing institutional challenges with sustainability in mind [1]. In the case of FSL, Fernández-Carvajal and Sequeira-Rovira [48] highlight the drive of women to transform the body they lead, and the implementation of new innovative projects. A manifestly similar situation can be seen in the study by Proctor [52], where leaders highlight the changes they have managed to introduce to the institution as their main source of motivation. Thus, they emphasise the achievement of new facilities and resources, or the resolution of some of the problems faced by the organisation, leading them to obtain a high level of professional wellbeing. The value of learning and experience tends also to be highly rewarding, as it allows women to make themselves known within their organisations, especially given the scarce opportunities that female academics usually find to participate in male networks of influence [53]. Indeed, according to the results of Fernández-Carvajal and Sequeira-Rovira [48], the exercise of a managerial function, and the greater visibility that comes with it, offers female leaders the possibility of promotion and advancement to other positions of higher rank.

The contributions of women's leadership to the university contrast with what this institution offers women leaders, for, despite its claim of being an institution based on equality, in reality, it is strongly influenced by stereotypical disciplinary cultures [54]. On this basis, it is possible to state that the university is still the focus of a gendered scientific culture. This academic culture or tribe [55] reflects a set of norms and values developed and internalised over time that govern the way its members interact. To help solve issues of inequity in the STEM and academic leadership fields, in 2001, the National Science Foundation (NSF) launched ADVANCE IT [56], an institutional transformation programme initially consisting of nine US universities. The goal of the programme was to increase the participation of women in science and engineering (S&E) and to promote their full participation at all levels of academic administration, particularly in leadership positions, through the transformation of institutional practices, policies, and culture. The purpose of ADVANCE was to increase both the percentage of women in senior faculty positions, such as tenure and professorships [57], and to help achieve gender equity in STEM with professional development grants for women scientists in engineering [58,59]. In subsequent years, ADVANCE was expanded to 37 universities [60]. The rationale for ADVANCE is based on the growing recognition that the lack of women's participation at senior levels of academia is a systematic consequence of academic culture [61]. Thus, while the NSF specifically notes the importance of recruitment, retention, and advancement of women in STEM fields in changing the academic culture, a process of institutional transformation is needed. This process involves not only changing day-to-day actions, but organisational culture, customs, norms, communication style, management, and ways of thinking. In sum, ADVANCE, while seeking changes in the number of women in STEM fields of representation, addresses the issue of equity through deeper institutional change.

### 1.4. Influence of Gender on Sustainable Leadership Satisfactions and Opportunities

The debate on women's leadership style has attracted increasing interest in academia [62–69]. Traditionally, leadership has focussed on the role of a leader who, in most cases, is a 'great man' [70,71]. In this 'heroic, masculine' conception, the leader bears all the burdens and works hard to achieve goals. This approach assumes that leaders

are born to lead and the rest to follow. It seems this model is still valid [72], as organisations, in general, are still a long way from valuing women's leadership [73].

Leal-Filho et al. [3] conducted a study with 50 leaders from different universities around the world. The findings show that the problem of underrepresentation of women in leadership positions is persistent in university education. Thus, only 36% of the respondents indicated that women held more than 30% of the positions. In addition, respondents were not aware of gender-related issues and related actions that are necessary to achieve sustainability. When asked whether women were more effective sustainable leaders than men because of their greater concern for sustainability, almost half of the respondents (44%) remained neutral. However, 68% of participants noted that there is a greater focus by women on designing and implementing sustainability. These findings are not isolated, as they have been corroborated by other studies [62,69,74,75], some of which show that women in leadership positions consider their experience as leaders to be based on their sacrifice and hard work [76].

The satisfaction, opportunities, and contributions of women academics to the university would not be possible without a regulatory framework that promotes equality between men and women. For this reason, Spain has developed a set of laws and policies that seek to guarantee the right to gender equality in higher education institutions. Among them is the Spanish Organic Act 1/2004 of 28 December on Integrated Protection Measures against Gender Violence [77]. This legislation aims to provide a complete and comprehensive response to the different types of violence against women. In addition, it contemplates the implementation of a set of actions in the educational sphere to promote gender sensitization and awareness.

It is also important to mention the Spanish Organic Law 3/2007, of 22 March, for the effective equality of men and women [78]. This law pays special attention to the eradication of inequalities experienced by women in the professional environment and in labour relations. To this end, it introduces some key concepts such as parity, the reconciliation of personal, family, and working life, and the promotion of co-responsibility. It also recognises the principle of balanced presence in positions of responsibility. This is intended to guarantee the equal representation of women and men in these positions. Based on this, the Spanish Organic Law 4/2007, of 12 April, which modifies Organic Law 6/2001, of 21 December, on universities (LOMLOU) [79] raises the importance of contributing to the achievement of real equality between women and men. According to this law, the academy must not only assume the principles of tolerance, equity, and equality as an essential part of its aims and activities, but must also establish a series of mechanisms to guarantee the equal composition of representative bodies and the promotion of female participation in research groups. Similarly, it is envisaged that university authorities should remove possible obstacles and barriers that limit the presence of women in governing bodies and at the highest levels of the academic career. Special mention should also be made of the Royal Decree 1401/2018, of 23 November, which creates the Women, Science and Innovation Observatory (OMCI) in the Spanish science, technology, and innovation system [80]. Its main objective is to evaluate and diagnose the effectiveness of the policies implemented to increase the presence of women in the scientific field.

Despite the existence of the aforementioned policies, gender equality is still far from being achieved in Spain. While they have shown their potential to reduce gender inequalities, they have not succeeded in dismantling university power hierarchies [81]. To overcome the obstacles, barriers, and challenges to achieving gender equality in academic leadership, it is necessary to go beyond the legislative approach. This means implementing practical approaches that include equal treatment, positive action, and effective gender mainstreaming [82]. Equal treatment means eradicating the idea that women can succeed in academic leadership as long as they behave like men. Affirmative action measures are based on the recognition that there are differences between men and women and that a level playing field must therefore be created for both men and women. Finally, gender

mainstreaming can be described as the promotion of gender equality in the institutional structure, policies, and ways of seeing and doing in the university.

While the academic literature has addressed a people-oriented model of women's leadership [83] and the limitations women face in accessing positions of power in academia, there is a gap in the research on the potential for sustainable women's leadership, characterised as holistic, technical, and innovative, with ethical values and principles. These capabilities could enable women leaders to guide organisations towards a more sustainable state in the face of social, economic, and environmental complexity and conflict [16]. Equally few qualitative studies have investigated the impact of gender on the benefits, contributions, and opportunities of sustainable women's leadership for both the university institution and for women themselves. Against this backdrop, the objectives of this study are (1) to identify and analyse the satisfactions, opportunities, and contributions of female academics to university governance, and (2) to learn about their perception of the potential impact of gender on these possible outcomes.

## 2. Materials and Methods

In order to achieve the stated aims, the study was designed using a qualitative research approach. The use of the latter is justified by the concern to understand how leaders interpret and make sense of their particular reality [84]. Another reason for its choice is the rebalancing of power and control in favour of greater horizontality and equality between the researcher and the phenomenon under study [85]. Furthermore, the qualitative tradition is particularly suited to penetrate the experiential and to understand the richness of nuances of the socially constructed reality of the leaders [86].

### 2.1. Participants and Context

The study included 48 female academics from the University of Alicante (UA) (Spain) who had headed academic governing bodies. Of these, 50% were between 51 and 60 years old, 31.25% were between 41 and 50 years old, and 16.67% were over 60 years old. Only one of the female academics (2.08%) was between 31 and 40 years old. In terms of length of service at the university, 52.08% had professional experience ranging from 21 to 30 years, and 16.67% had been working at the UA for more than 30 years. Regarding the management responsibility of the interviewees, 54.16% of the participants had been in management positions in a university department. The remaining 45.84% were distributed among those who had assumed a deanship or the direction of a centre (16.67%), a research institute (16.67%), or a vice-rectorate (12.50%). Most of the participants (60.41%) had a seniority ranging from 0 to 4 years. Only 33.34% of the women leadership academics had experience of between 5 and 8 years. This implies that few women persist in leadership after a first term.

The UA is geographically located in Alicante, a Spanish province in the southeast of the Iberian Peninsula and part of the Valencian Community. It is a public institution of higher education with an international projection, whose mission is the integral education of students [87]. This necessarily entails respecting, defending, and promoting the right to effective equality between women and men in all the dimensions and practices of the institution. During the 2019–2020 academic year, the number of students enrolled in its official studies (Bachelor's, Master's and Ph.D.) was 25,635, and 3941 people worked at the university, including administrative staff (1383) and teaching and research staff (2558) [88]. Currently, this higher education centre offers a wide range of courses, including 54 official undergraduate studies, 62 postgraduate and master's degrees, and 31 doctoral programmes, 87 specialisation degrees, and a wide range of complementary training courses, especially in languages and languages.

This institution is firmly committed to the promotion of gender equality and the integration of sustainable development in all its spheres of action (teaching, research, and knowledge transfer). It currently has a wide range of instruments and strategies to raise the awareness of equality and sustainability among the university community. With regard

to the empowerment of women, the UA has been able to provide itself with a broad regulatory framework that aims to guarantee the balanced composition of men and women in management teams [89]. It also has a Vice-Rector's Office for Social Responsibility, Inclusion and Equality, which designs and implements policies linked to sustainable development and the promotion of the role of women in academic life (https://bit.ly/3MuCtmd, accessed on 14 January 2022). In the field of research, it has the University Institute for Gender Studies Research, an interdisciplinary centre dedicated to the promotion of scientific research in the field of equality (https://ieg.ua.es/, accessed on 11 February 2022). Furthermore, in order to promote gender mainstreaming in teaching and to guide the governing bodies, services, and units on the principle of equality, the UA has an Equality Unit (https://web.ua.es/es/unidad-igualdad/, accessed on 25 March 2022), an Observatory for Equality between Women and Men (https://bit.ly/3O3ej3d, accessed on 27 March 2022), and four equal opportunities plans, the latest of which was recently published [90]. With regard to its commitment to the 2030 Agenda, its full integration into the institutional Strategic Plan (2022–2024) [91] should be highlighted. In practice, this is reflected in actions as diverse as the establishment of agreements with other organisations in the area of sustainability, the organisation of training courses on sustainable development, the announcement of the master's degree awards to stimulate research on the Sustainable Development Goals (SDGs), and awareness campaigns for more sustainable campus management. Based on the characteristics of the context, it is considered that the study of the satisfactions, opportunities, and contributions of academic leaders to SL may be of particular interest to other realities.

### 2.2. Instruments

We chose to employ the semi-structured interview [92,93] as the data collection technique. This has been particularly valuable for leaders to relate their life experiences around SL performance because of its situated nature and its suitability for making veiled aspects of human and organisational behaviour visible [94,95]. The interview script consisted of 18 questions. In this study, we analyse those that focussed on inquiring into the satisfactions, opportunities, contributions, and gender of female academics (four questions):

- What satisfactions have you found in your performance as a leader?;
- What opportunities have you had in your performance as a leader?;
- What do you think you contribute to the institution through your work as a leader?;
- Do you think your gender has had an influence on these issues?

In order to contextualise and deepen the issues investigated, the following sociodemographic data were added to the instrument: age of the participants, period of professional relationship, management position held, and professional experience in the position.

### 2.3. Procedure

Initially, the academics were contacted by e-mail. They were informed of the objectives of the study, and of the anonymous, voluntary, and confidential nature of their participation. Subsequently, a schedule of available dates was created according to their availability. The data collection process was carried out over a period of four and a half months. The interviews were conducted orally and were audio-recorded. At the beginning of each interview, participants were reminded of the purpose of the study in order to obtain informed consent. In addition, it was emphasised that, as prescribed by the ethical principles of research defined in the Declaration of Helsinki, all data provided would be confidential and anonymous. After the collection of the information, an initial content analysis process was initiated to check the validity of the information, and to establish the first connections between the emerging concepts.

### 2.4. Data Analysis

The interviews were transcribed into separate documents using the Microsoft Word 2010 word processor software (Microsoft Corporation, Washington, DC, USA). Given the

cyclical and reflexive nature of qualitative research, its flexibility, and the segmentation of the data into units of meaning [96], a first draft of codes was made based on the four questions that made up the interview.

In the first phase, the starting point was the reduction of the data by segmenting the text and coding with repeated variations. To do this, the text was classified into parts according to the characteristics of the phenomenon under study, and a code was established for each significant unit. The categorisation process was based on intensive and repeated reading to improve the comprehension of the content and, consequently, to distinguish the emerging units of meaning. In order to carry out this coding, a practice that was useful during the first readings of the transcripts was the constant questioning of the text. Such questions were particularly suitable for formulating codes that were not only descriptive, but also explanatory and interpretative in nature. Another substantially valuable element was the description and exemplification of each of the emerging codes. The organisation and representation of the data was then addressed through the use of different double-entry matrices. These were particularly beneficial for a visual understanding of the ordering of the phenomena, the identification of cause–effect relationships, and the connection between the different categories. On this basis, and with the knowledge accumulated through the analysis of the narratives, the initial outline of codes was modified, completed, and validated by the three experts in gender and management, and by two researchers who were specialised in a qualitative methodology. The systems of analysis followed were the retrieval of the coded text and the study of frequencies. The former allowed access in an ordered way to sections of the text that share the same label, i.e., that are related to each other. In addition to the analysis of the coded text, a frequency study was carried out. This was used to collect the aspects most emphasised by the participants in their narratives. Specifically, a content analysis of conventional and summative information was carried out [97]. Each of the female academics interviewed was assigned an alphanumeric coding (ACA_XX), representing the participant's number. Subsequently, the data were processed with the Aquad 7 software [98]. This program made it possible to categorise and organise the data, and to draw conclusions through the relationship of categories.

## 3. Results

Table 1 presents the results of the absolute frequencies (AF) and the percentage of these (%FA), where AF is the number of times participants allude to a unit of meaning and %AF is the ratio of that code to the total AF (AF × 100/total AF). In addition, the presentation of the qualitative results is supported by the narratives extracted from the participants' voices, which illustrate the meaning of each of the emerging codes.

The structuring of the inferred codes responds to the desire to know, not only the satisfactory experiences that the leaders find in the development of their function, but also their perception of the impact that gender may have had on them. Table 1 shows the occurrence of these significant units in their speeches.

The issues most highlighted by the participants were: types of professional (AF = 90; %AF = 22.73) and personal (AF = 44; %AF = 11.11) satisfactions and opportunities, such as the contribution to the institution consistent with their personal qualities and values (AF = 56; %AF = 14.14), and the extension of networks (%AF = 46; %AF = 11.61). In contrast, subcode 2.3., 'No opportunities', had a smaller number of cases (AF = 10; %AF = 2.52).

**Table 1.** Codes and sub-codes for satisfactions, opportunities, contributions, and gender.

| Category | Codes | AF | %AF |
|---|---|---|---|
| | 1. Types of satisfactions | | |
| | 1.1. Personal | 44 | 11.11 |
| | 1.2. Professional | 90 | 22.73 |
| | 1.3. No satisfaction | 15 | 3.80 |
| | 2. Types of opportunities | | |
| | 2.1. Learning | 29 | 7.32 |
| | 2.2. Expanding networks | 46 | 11.61 |
| Satisfactions, opportunities, | 2.3. No opportunities | 10 | 2.52 |
| contributions, and gender | 3. Type of contribution to the institution | | |
| | 3.1. Boosting staff | 9 | 2.27 |
| | 3.2. Experience and knowledge | 16 | 4.04 |
| | 3.3. Network building | 19 | 4.80 |
| | 3.4. Personal qualities and values | 56 | 14.14 |
| | 4. Perception of the conditioning influence of gender | | |
| | 4.1. Neutral perception | 31 | 7.83 |
| | 4.2 Positive perception | 31 | 7.83 |
| | Totals | 396 | 100 |

AF = Absolute frequency; %AF Absolute frequency percentage.

### 3.1. Code 1. Type of Satisfaction

One of the main elements that seems to influence the attitude towards professional responsibilities is the degree of satisfaction that a person finds in the performance of their function. This, in turn, is particularly significant for the commitment and motivation felt in the task to be performed. The interviewees feel proud of their trajectories as leaders and allude to satisfaction of a professional nature (and, to a lesser extent, personal satisfaction). Lastly, there are some stories which, although not very frequent, are indicative of the disenchantment that leaders seem to experience with the exercise of their managerial function.

#### 3.1.1. Subcode 1.1. Personal

In this subcode, we have categorised the narratives alluding to the socio-emotional enjoyment that the leaders find in academic governance. Although this type of satisfaction is not the most reiterated by the interviewees, it has a significant presence in their discourses (%AF = 11.11). In inferring the semantic nuclei referring to these satisfactions, it is possible to highlight a series of nuances derived from the causes of them. Thus, the participants insist on the personal enjoyment they derive from the recognition, respect, and support of the people they lead. As one interviewee stated,

> Well, both in the positions I have held in the direction of the school and the department, the truth is that you have many satisfactions. Because first, if you treat people well and respect people, then people respect you a lot. What you give, you receive, and that is nice (Aca_19).

One should not lose sight of the need that some women traditionally have to be recognised and valued by others:

> So, people value that a lot, and say to you, man, the first woman! You think it was not because of that, but they kind of value it above other things, and then you feel satisfied (Aca_26).

On the other hand, some leaders emphasise the achievements they have been able to accomplish through their own effort and personal sacrifice. That is, they focus on the self-improvement aspect of a personal challenge, especially when they see that, despite their initial doubts and insecurities, they have been able to face the challenge successfully:

> Then, I have managed to overcome obstacles that I thought were insurmountable on a personal level, also when it comes to relating to people and all that, for me it is very satisfying (Aca_44).

Leaving these kinds of barriers behind and becoming aware of their true leadership competencies ultimately leads them to experience a deep sense of personal fulfilment:

> But at the same time, as things have been going well for me since then, it has paid off and I have had a good and important sense of personal fulfilment that has been worthwhile (Aca_26).

### 3.1.2. Subcode 1.2. Professionals

The reason for the enthusiasm experienced by the participants lies in the transformation they have been able to implement in the organisation they lead. The high level of insistence with which they allude to this issue in their interviews (%AF = 22.73) is evidence that this type of achievement becomes, in practice, the main source of satisfaction. Thus, they emphasise the optimisation of the work climate and the cohesion of the group they lead. One interviewee puts it this way:

> In other words, I am happy because I think that when I left management, I left a department that was more united than it was initially, I think. So, in that sense I am very satisfied (Aca_34).

The participants also highlight the improvements they have achieved in terms of equipment, both material and in relation to economic resources:

> Before we had no money, no computers. Now we have a good library. To be able to do in this sense, and to be able to ensure that those who now enter the department have a better infrastructure . . . that gives me great satisfaction (Aca_04).

Some of them feel that their work has promoted the recognition and presence of the organisation within the campus and also outside it:

> One enormous satisfaction was that this centre has become a reference point outside the university (Aca_28).

> Well, it was satisfying to see that the department was growing, that everything was working well, that we were a model at that time, which is not the case now. We were a model in the university, a model of [a] department, of organisation, of academic results, of research results. I think everything was going very well (Aca_40).

Moreover, they especially highlight the work of a whole team.

> Well, we have been able to create an institute that is doing very well, that is working, [where] . . . there is a powerful team of people with very good synergy, and I think that this is the reading that is made at the end (Aca_22).

The training of the teaching staff and the opportunity to have an outstanding teaching staff is also an important source of satisfaction.

> And then the experience of the part-time teaching staff is also very satisfactory, which constantly connects us with reality, and, in addition, I am lucky to have a very high [level of] qualification in all of them, in all of them. Yes, yes, true (Aca_08).

However, as far as teaching is concerned, the main source of the leaders' pride seems to lie in the improvements they have implemented in the design of the degrees:

> When I took office, the centre had a three-year degree, and students could not go beyond the bachelor's degree. At the end of my term of office, the centre had two degrees, [including] a master's degree, [and] the students could already do a doctorate (Aca_12).

Finally, some participants emphasise, albeit with a lesser degree of insistence, the satisfaction they experience as a result of the improvement of research. In this context, they

emphasise above all their work in promoting and strengthening the research activity of the body they lead, thus positioning it in the international arena.

> The first, then, is to be the most cited department in the Ibero-American area. Of the researchers in this area, we are the most cited in Ibero-America. Maintaining this position is important and very satisfactory (Aca_08).

### 3.1.3. Subcode 1.3. No Satisfaction

A group of participants claim that they do not feel attracted to university governance and that, consequently, they do not find it a pleasant task (%AF = 3.80):

> Man ... I have already told you that it was not my desire to be a director either, because I don't like it, I don't like it, I don't like to command, I don't like to be in positions like that (Aca_48).

These women took the position because the circumstances required it; however, they admit that they do not feel attracted to this professional field.

> The truth is that I didn't feel like it, I don't like being a department director. No, but well, you have to accept it because it was the right thing to do (Aca_06).

> But I have not applied myself because I do not like university management, of course. I would not like to be a rector or vice-rector or anything like that (Aca_05).

### 3.2. Code 2. Type of Opportunities

One of the main opportunities leadership seems to provide for women is the possibility of access to a wider circle of people, information, and resources. Indeed, in subcode 2.2., 'Expanding networks' is the most prominent. It is important to underline that the exercise of leadership also seems to represent, in practice, a particularly valuable learning experience. With a lesser presence, those allusions to not enjoying any kind of advantage are noticed.

### 3.2.1. Subcode 2.1. Learning

The performance of a professional task usually entails the acquisition of new knowledge and skills, especially in a field with such disparate and diversified profiles and conditions as academia. In this context, SL can become an extraordinary learning opportunity for female academics, who are usually entering a space of absolute novelty for them. This subcode accumulates 7.32% of absolute frequency, from which it can be inferred that academic governance constitutes a magnificent learning path for leaders:

> Well, I think that yes, maybe at the first moment you don't realise it, but you realise that there are things to learn, that you learn from it. When a little time has passed, you realise that there are very important things that you learn (Aca_14).

### 3.2.2. Subcode 2.2. Expanding Networks

The women academics argue that leadership has been useful for increasing and expanding the networks they have in the institution. As a result of their positions, they have been able to access people, resources, and information that they otherwise would not have been able to. This seems to be the main benefit that the leaders have found in academic governance (%AF = 11.61). Thus, it is possible to note constant mentions in their accounts of the possibility that the position gives them to meet and interact with new people within the institution itself.

> Man, maybe, look, what it did allow me to do was to meet more people, because there were many directors of other institutes, both in the arts and sciences, that I did not know (Aca_10).

However, this broadening of the network of contacts is not limited to the university itself, but transcends it, and goes much further. Thanks to their managerial roles, some of the leaders are able to establish new links with professionals and scientific collaboration with other institutions.

> But on the other hand, I have met more people that we could collaborate [with] in research tasks, and in fact, things have come out of there, more multidisciplinary works, with points of view that can be more interesting (Aca_29).

In turn, this greater prominence within the institution has opened certain doors for them much more easily and quickly:

> As for those above me, I do have more access. So when I think it is convenient, I say who I am and ask to speak to whoever it is, and what I notice is that they give me appointments much more quickly than before (Aca_36).

Moreover, it is possible to identify a set of voices that are better enabled in their knowledge of everything that happens in the institution:

> Then, well, hey you don't stop being part of the Governing Board, you don't stop having interviews with people who are at the level, let's say broader management, ... [and] you [therefore] have better information of what is happening (Aca_17).

On other occasions, the knowledge provided by their advantageous position is used for more effective decision-making and for the promotion of the body they lead.

> On the one hand, you can be in the places where there is more information, know what is going on, and that also helps you to make decisions or, at least, know what is happening in the university (Aca_32).

### 3.2.3. Subcode 2.3. No Opportunity

There are also voices that recognise that they do not find any type of benefit associated with leadership (%AF = 2.52). At this point, it is important to remember the feminine tendency to repel any formula of extra benefit at the work level for fear of rejection. Therefore, the incidence of this topic in their interviews could be conditioned by that social image they wish to avoid:

> It is that, let's see. I never see any post or position that I have to perform as a platform or opportunity for other things. I just don't see it, I don't understand it that way (Aca_01).

In fact, their determination not to be identified as beneficiaries of some kind of prerogative leads some of the participants to insistently deny the enjoyment of, or desire for, additional opportunities.

In an exercise of professional honesty, they make every effort to disassociate themselves from those who, in their opinion, are kept in governance by some kind of interest.

### 3.3. Code 3. Type of Contribution to the Institution

The study of SL necessarily involves the analysis of those elements with which the participants contribute to the development and growth of the institution. Through this exercise, it is possible to identify how the women evaluate their own contribution, and what the contributions are with which they signify their work. In this sense, the main concessions that the narrators make to the organisation are located, above all, around their personal attributes. Subcode 3.4., 'Personal qualities and values', accumulates the highest number of versions of expression. In second place, and with an ostensibly significant difference, are the allusions to network building and to experience and knowledge. With the lowest absolute frequency rate, interviewees refer to the help and guidance offered to the members of the organisation they lead.

### 3.3.1. Subcode 3.1. Encouraging Staff

One of the effects that has traditionally been attributed to female leadership, inside and outside academia, is the help and guidance given to the members of the group they lead. Thus, women in management positions in the university are considered to have a special concern for promoting the growth and professional advancement of their staff.

However, the number of occurrences of this subcode is the smallest, with a 2.27 absolute frequency. From this, it can be inferred that either the interviewees are not fully aware of the motivation they give their team members, or that motivating staff is not among their main contributions. Even so, a number of voices emphasise their contributions to boosting the research activity of their group members:

> And to see how people who started as assistants, who are doctoral assistants, who are hired as doctors, who are in the process of being accredited for a tenured position, and to see how you have been able, in some way, to help this trajectory (Aca_08).

Closely related to this type of contribution is the achievement of new positions for colleagues who are in a position to be promoted.

> My contribution? For example, getting a position [of] assistant doctor for a person who had been asking for it for years and had not been granted it. It is true that times have changed, and things are better, at least in the vice-rectorate or rectorate. But it is true that for years we had no vacancies for young people, and we got one. Not as many as we wanted, but listen, we got one (Aca_16).

Likewise, it is necessary to emphasise that female academic managers' determination to favour the encouragement of others even leads them to support people who do not belong to their group:

> I also encouraged all colleagues, even those not in my team; for example, I supported and helped them all study for a master's degree (Aca_43).

### 3.3.2. Subcode 3.2. Experience and Knowledge

This refers to the textual segments that circumscribe the participants' contribution to the knowledge they have accumulated throughout their professional careers in different areas. Their narratives show that this type of contribution is not one of the main concessions that the leaders make to the institution (%AF = 4.04). In the first place, testimonies are identified that insist that their most significant contribution to the institution is due to the previous knowledge and experience they have gained in the field of leadership.

> I think that covering, suggesting certain possibilities in this line that I am talking about, of a little bit of promoting research activities, which I also have experience as director of an R&D project (Aca_13).

The narratives that highlight the value of the knowledge they have of the University of Alicante also stand out. These women have had a long career in the institution. Therefore, they are expert connoisseurs of its culture and its operating mechanisms.

> I have known the University of Alicante since its beginnings, [and] since ... 1980 when I started studying there. So, I bring a very useful knowledge of the university (Aca_26).

Other testimonies emphasise, in a specific way, the interviewees' contributions to the field of research and teaching. The participants highlight their trajectory in these areas, and state that, thanks to this, they can adopt a broader vision in the exercise of their function, as well as taking into account other aspects, or even better contributing to the promotion of the institution.

> Well, I believe that experience, because I had many years of teaching and professional experience as well, with experience in research. So, I think this was also important because it was a broader vision (Aca_32).

### 3.3.3. Subcode 3.3. Network Building

The possibility of establishing new links and contacts with other institutions is also among the main contributions adduced by the participants. The findings show a low presence of contributions corresponding to this coding (%AF = 4.8). Despite this, a number

of testimonies allude to the leaders' efforts to reinforce the feeling of unity within the team itself, their contributions to achieving a common goal, the sum of efforts, and the creation of synergy to stimulate coordinated action.

> And above all, try to involve. I believe that we have to involve people a lot. We made a great effort to get everyone to go hand in hand. I think it was important (Aca_23).

The main contribution of the participants is also the linkage with other agencies within the institution. The leaders have a greater degree of openness and a conviction that this type of contact allows the organisation to move forward.

> Then I also try to maintain close relations with departments equivalent to my own, so that this is also good for our faculty (Aca_33).

Likewise, the leader facilitates the possibility of establishing new relationships with other entities and structures in society. This allows a greater transfer of knowledge:

> But I have also managed to get many people from outside the university to join. That is to say, we cannot underestimate either. On the contrary, all the signs of support . . . we have been able to gather. Of course, that is obvious (Aca_18).

### 3.3.4. Subcode 3.4. Personal Qualities and Values

This covers all the narrative segments alluding to the characteristics and principles by which their work has been distinguished, according to the participants. The analysis of their testimonies shows that this type of contribution is the most reiterated by the leaders (%AF = 14.14). Therefore, when it comes to estimating their most significant contributions, they emphasise attributes of a personal nature. However, sometimes, their adherence to the stereotypical image of modest and humble women is such that it is difficult for them to recognise this type of contribution.

> Well, I don't know. I don't know if I contributed. I don't know. Maybe that's for someone else to say, right? Me contribute? I don't know, I think that the way of management was different, wasn't it? And maybe [there's] a little bit of nuance . . . in the management [that] may be more my personal stamp (Aca_10).

Likewise, there are stories that emphasise the contribution of those qualities that favour the improvement of the work environment, among which the capacity to empathise with others stands out.

> Well, look, now talking to you, I believe that empathy. That is, I believe that things are achieved not so much with force or belligerence, but with the ability to empathise with others, and that is what I think I contributed (Aca_35).

There are also frequent allusions to their capacity for listening and dialogue—qualities with which they claim to be able to create an atmosphere in the organisation characterised by trust, closeness, and the understanding of others. As one interviewee puts it,

> So, I believe that I am a good listener. I know how to listen, and I know how to accept that your opinions or decisions are not accepted by everyone (Aca_07).

Some highlight their negotiation skills, as well as the achievement of shared understanding among all members of the group. Once again, however, the difficulties they seem to have in valuing their contributions are noticeable:

> Well, I couldn't tell you what my personal touch was either, because I wasn't aware that I was imprinting any personal touch on anything, was I? I think that my touch was a little bit the consensus, and on the other hand it was also the dedication, being very aware of things (Aca_41).

This is in addition to believing that emotions and affection also play a transcendental role for the work to flow and for the team to grow beyond individualities.

> And then, I believe that affection also plays a role, eh? Because here emotions and feelings also count (Aca_16).

In contrast to this type of contribution, those stories that emphasise the principles and values with which the female academic managers contribute to the sustainable development of the organisation are also noteworthy. In this context, their statements of the responsibility and professional ethics with which they claim to carry out their work are particularly significant.

> Well, not much, but basically something important. I believe that if each one, in the scope of their responsibility, has an ethical requirement with their work, is concerned about moving the university forward, as a sustainable university, in all aspects, "another rooster would sing for us in general" (Aca_04).

Some of the leaders emphasise their commitment and dedication to their work, especially the time invested in the development of their function.

> I contributed many hours, many days, many weekends, vacations, and a lot of work in general. I strive to transform and improve the functioning of the university, and I believe that I can and must do this very well (Aca_04).

In addition, they emphasise their deep sense of justice and impartiality, temperance, serenity, and solidarity. According to them, these qualities allow them to face complex situations in the performance of leadership.

> If you only ask me about the direction of the department, what I brought was, I believe, fairness. That is to say, a fairer, more supportive operation and that there was no nepotism (Aca_30).

> I don't know, maybe serenity and temperance when things get a bit ugly, that's what I contributed (Aca_37).

These qualities are in addition to the transparency with which they claim to face possible disagreements with colleagues.

> I liked it very much . . . I think we are in a different era now, for example. I liked to inform everyone about every step that was taken, everything that was done, everything that happened, so that everyone was informed, could participate, and could decide whether to enter or leave, to get involved. I think that's what I contributed the most (Aca_45).

*3.4. Code 4. Perception of the Conditioning Influence of Gender*

In the analysis of this code, it is possible to know the assessment that women academics make of the influence of gender regarding their satisfactions, opportunities, and contributions and, consequently, the degree of critical awareness they have in this regard. The study of the testimonies reveals the existence of two subcodes: 4.1., 'Neutral perception', which brings together all those narratives that reject the possible influence of gender on the participants' satisfactions, opportunities, and contributions, and code 4.2., 'Positive perception', which compiles the narratives that recognise the existence of a gender perspective among the satisfying experiences.

3.4.1. Subcode 4.1. Neutral Perception

Women academics consider that gender has not had any influence on the satisfactions or opportunities they find in the management function, nor on the contributions they have made to the organisation. This coding obtains 50% of absolute frequency. Those who adopt this position deny that gender could have mediated the satisfying experiences they have found in the exercise of leadership.

> As I have always been a woman, of course, since I have not been a man, I cannot compare. I don't know if others have the same satisfactions as me. But come on, I don't think there is any influence, I don't think so (Aca_12).

They believe that these have not been affected in any case by gender dynamics, but that any effects are due to personal issues, such as character or professional effectiveness:

> On opportunities? Well, I think that sometimes, more than gender, it's the way you are, your personality, your attitudes, the way you are, more than gender. Because I do have female colleagues who have handled it badly and male colleagues "who have developed rashes", that is to say, there is a bit of everything. For me it is a little bit about how you are and how you manage it (Aca_04).

In the case of contributions, they do not seem to believe that these are in any sense mediated by gender.

> Well, I contribute my work and what I can humbly contribute, but I don't think it has to do with gender bias (Aca_22).

### 3.4.2. Subcode 4.2. Positive Perception

Leaders recognise the impact of gender on their satisfactions, opportunities, and contributions. In fact, 50% of the narratives point to the conditioning influence of gender. Therefore, it is possible to affirm that, a priori, a broad set of voices highlight gender as a strength for leadership. There are numerous accounts that distinguish between men's and women's ways of doing things when both occupy management positions.

> When there is a women's meeting, unless something happens, you go to the meeting and you want to finish the meeting, right? I go to a meeting to finish the meeting. Men, however, go slower. I think that this marks a way of conceiving work, of conceiving time, of conceiving everything (Aca_20).

This is especially the case with regard to the use of time and priorities at work, where the female leaders claim there are significant differences. While they seem to be more interested in networks of power and influence, they focus mainly on performing daily tasks and fulfilling daily commitments.

> In those moments the men, take advantage to continue doing politics and to talk, while I use that moment to advance in my work, to rest, to call my mother, to try to finish earlier because I want to leave earlier, because my mother is sick, etc. (Aca_39).

Furthermore, this participant also highlights her constant search for consensus in decision-making processes and her more integral and holistic perspective of problems:

> Surely, we women do more of that, we tend to analyse and look for more integral solutions, and not to solve something that is momentarily resolved (Aca_20).

Finally, in the same terms, participants refer to women's capacity for teamwork, which is, in their opinion, notably superior to that of their male counterparts.

> It is possible, I think, that maybe the ability of women to work as a team is much more developed than in the case of men (Aca_11).

## 4. Discussion and Conclusions

Concerning the three dimensions of SL identified by Visser and Courtice [1], it is possible to conclude that the leaders possess traits, skills, and knowledge, both general and specific, that are characteristic of this leadership. In this sense, they emphasise concern for the organisational culture and the achievement of objectives. Likewise, they develop inclusive, fair, and transparent leadership, and have a systemic understanding of the problems affecting their organisations. Their willingness to innovate, their ethical principles, and their innovative systemic thinking are remarkable. They have shown themselves to be visionary, creative, and forward-thinking, and they try to deal with complexity and conflicts, especially socio-economic ones, prospectively.

It should be noted that the participants' main source of pride is of a professional nature, namely, they feel satisfied with their contributions to the improvement of the

institution. In a way, they have the feeling of having fulfilled their duty, especially given that their main motivation for access is their deep commitment and desire to participate in institutional construction and development [3]. This type of satisfaction has also been highlighted in other similar studies, such as that of Sánchez Moreno and López-Yáñez [99] and Wroblewski [100], which illuminate the pleasure generated by female university leaders in becoming transformative agents and catalytic agents of the organisation. More specifically, they are particularly proud of having been able to improve the institutional climate, increase resources, and implement structural changes, as well as projecting and promoting the image of the body they represent, while trying to find a balance between institutional interests of an economic and social nature [101]. In short, their actions focus on developing the organisation by adopting socially responsible strategies [2].

On the other hand, monetary reward is not an encouraging factor with regard to the leadership of female academics. On the contrary, the participants point to the emotional motivations highlighted in the studies of Bianchi et al. [44] and Divya et al. [45]; these are exemplified by the relationships they have been able to forge, or the self-improvement of possible challenges. Ultimately, and although there is a majority of positive evaluations of the experience, there are also those who openly state their rejection of these types of activities. The reasons for this, they argue, lie in their disaffection with the managerial function, their identification of leadership with models of a more personalistic and individualistic nature, or their lack of vocation for this type of task. The study by Fernández-Carvajal and Sequeira-Rovira [48] ratifies this trend, stressing that the administrative nature of the function, and the political component, constitute some of the reasons that may lead women to show disinterest in these types of tasks.

However, this does not prevent women leaders from recognising the opportunities that the exercise of leadership has offered them. They recognise that they have enjoyed multiple and varied advantages, among which the possibility of widening the circle of people, information, and resources on which they have been able to rely stands out. Undoubtedly, occupying a position in the academic structure allows them to interact with others, both inside and outside the institution, to access first-hand information, and to learn more about the organisation. In fact, it could be asserted that the position of leadership becomes a privileged one, enabling the leader to have a more holistic view of the campus, and also to project oneself and make oneself known [42]. This possibility is especially positive and beneficial in the case of female academics, as they often have serious difficulties in accessing informal networks of support and influence [75,102–104], and, in many cases, they have a smaller circle of contacts than their male counterparts [105]. These social networks become, in turn, a fundamental resource for sharing professional obligations and for the ability to advance strategically [106,107], which ends up benefiting and favouring the career trajectory of female leaders.

On the other hand, and with less incidence in their interviews, there is also the possibility of learning. The participants' testimonies show that the performance of the position allows them to acquire valuable skills and abilities in a field that is practically unknown and new to them. These findings are concomitant with the studies of Rodríguez and Aguiar [43] and Rodríguez et al. [42]. Indeed, most of them are multifaceted women, with multiple concerns and a broad desire for knowledge, but with few previous opportunities for training in this area, therefore leadership ends up becoming an appropriate space to discover new learning. Some even recognise that a large part of their identity and professional trajectory is due to their experience in areas of responsibility. Therefore, it could be affirmed that although leaders contribute with their effort and work to institutional improvement, they are also rewarded in multiple ways and forms [52,95]. In the face of this type of valuation, there are also some voices that admit to not having found any type of benefit in academic leadership. A possible explanation for this trend lies in the influence on them of meritocratic politics. As Peterson [102] rightly points out, women leaders are often accused of having betrayed the principles of academic meritocracy, by having gained access supposedly because of their status as women, and not because of their ability. In this

sense, the results of our study show that holding the position represents an extraordinary opportunity for enrichment and professional growth for women leaders, as, in addition to feeling fully satisfied for contributing to the improvement of the organisation, they are also able to expand their network of contacts and acquire valuable knowledge that is useful to them, in that it informs them of how to manage in the institution.

Concerning the contributions they make with their work, a dominant tone in the narratives is the humility and modesty with which they allude to their contributions—a behaviour that may be a reflection of the traditional gender roles in which they have been socialised, as well as the exacerbated demand and perfectionism that women tend to apply to themselves [108]. Nevertheless, regardless of this and drawing on their personal qualities and values, the participants value their contributions to transformation and the progress of the institution [51]. Among other aspects, they highlight those skills of SL related to the improvement of the climate of coexistence in the organisation, such as their capacity for empathy, listening and dialogue, their skills for negotiation, mediation, and conflict resolution, or their competencies for emotional management. They also emphasise the principles and values with which they clothe their role, such as professional ethics [12], their deep sense of responsibility, their commitment and dedication to work, loyalty to the institution, and fairness and impartiality [1,15,109].

Other contributions of FSL are the construction of networks, both within the body itself and beyond the institution, as well as experience and knowledge in management; these results are consistent with the study by Fernández-Carvajal and Sequeira-Rovira [48]. One of the main characteristics traditionally linked to female leaders is their concern for people [45]; however, the participants are not yet fully aware of their contributions in this area.

Regarding the perception of the impact of gender on the satisfactions, opportunities, and contributions they make to the institution, the leaders do not express a clear or decisive position. On the contrary, both the neutral vision and the positive perception of gender show the same degree of iteration in their speech. Thus, while there are testimonies that reject that their feminine condition could have affected their positive experiences in any way, the same number of narratives consider it to be a determining element. In fact, those who adopt this positioning tend to clothe the female leadership style with a series of distinctive features which are opposed to male ways of doing things [52,110]. Among other characteristics, these include a more efficient use of time, a greater degree of task orientation, low interest in networks of power and influence, more noticeable mastery of multitasking, and a superior capacity for empathy and teamwork—qualities that, in some studies, have been associated with female leadership [111]. However, it cannot be ignored that these types of beliefs may be the result of socialisation in traditional gender roles, which lead to the attribution of a series of differentiated traits to men and women [3,108,112].

In summary, beyond the satisfactions of a personal nature that female leaders may find, the main reward derived from the performance of these positions of management lies in the changes and improvements they have achieved for the institution, thanks to female academic leaders' effort, work, and dedication. Access to positions of responsibility has allowed them, first and foremost, to leave their own habitat, and gain visibility, thus building new networks or expanding existing ones both on and off campus. Their main contribution to the organisation is at the axiological level, as they are capable of providing the institution with a series of values and attributes aimed at transforming the university.

The research shows a model of women's leadership centred on learning and personal and social values, as well as a clear overall picture of women's leadership strengths, weaknesses, and opportunities. Nevertheless, while women academics have opened the way towards empowerment under the SDG target 5c [113], they still face personal and contextual challenges, such as their own insecurity, the lack of political priorities [114] and reward structures, and ingrained prejudices in university organisational practices. Despite the small scope of our study, the findings offer implications for policymakers and managers who should push for the greater participation of women in university governance, as their

input could be instrumental in advancing the sustainable development environment of the institution. The study can be useful for proposing practical leadership policies based on equal treatment and could be replicated in different contexts in order to contribute to a better understanding of the leadership of women academics in Higher Education.

**Author Contributions:** Conceptualization, G.M.-M., M.U.-S. and D.G.-M.; Data curation, M.U.-S. and S.B.-M.; Formal analysis, M.U.-S. and D.G.-M.; Funding acquisition, G.M.-M.; Investigation, G.M.-M. and M.U.-S.; Methodology, G.M.-M., M.U.-S. and S.B.-M.; Project administration, M.U.-S.; Resources, M.U.-S. and D.G.-M.; Software, S.B.-M. and D.G.-M.; Supervision, G.M.-M.; Validation, S.B.-M. and D.G.-M.; Visualization, D.G.-M.; Writing—original draft, G.M.-M. and M.U.-S.; Writing—review & editing, G.M.-M., M.U.-S., S.B.-M. and D.G.-M. All authors have read and agreed to the published version of the manuscript.

**Funding:** This research was financed by the University of Alicante and the Generalitat Valenciana (Spain) in the framework of the project "Research in Education for Sustainable Development: Integrating the 2030 Agenda in the university classroom. Honduras/Nicaragua/El Salvador" (RES. CUD-09-26/2022).

**Institutional Review Board Statement:** Not applicable.

**Informed Consent Statement:** The data are not publicly available due to confidentiality agreements with the participants and IRB requirements. Any questions can be directed to the first author.

**Data Availability Statement:** Not applicable.

**Conflicts of Interest:** The authors declare no conflict of interest. This paper is the result of the doctoral thesis "El liderazgo de las académicas en la Educación Superior: un estudio de caso", presented by Mayra Urrea-Solano, in January 2021, to opt for the title of Doctor in Educational Research, within the Doctoral Program in Educational Research at the University of Alicante (Spain).

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
