# Peer review of "The Satisfactions, Contributions, and Opportunities of Women Academics in the Framework of Sustainable Leadership: A Case Study"

_sustainability, doi:10.3390/su14148937_

Round 1

Reviewer 1 Report

This paper examines sustainable leadership prospects for women academics. The following changes would improve the paper.

1. Is it LS or SL? Both are used.

2. There is a strong theoretical literature on gendered organizations/institutions and academia as fitting this set of characteristics. The National Science Foundation has funded a lot of this work concerning STEM professors. This research could be cited with greater attention to its linkages to the present investigation. 

3. Please speak more persuasively about the value-added dimensions of using qualitative methods in this project. What do they do that cannot be achieved by a survey?

4. Please make a better case for Spain as a site for this research. What can we learn from this particular case that is especially valuable?

5. The instrument is the questionnaire. Semi-structured interviewing is an approach to qualitative interviews.

Author Response

Dear reviewer, thank you very much for your contributions which have undoubtedly improved our work substantially.

Please find attached a document responding to all your comments. 

Thank you

Reviewer 2 Report

This is an interesting article,  well written and complete. The objectives of the article as posited in the beginning are achieved.  I have therefore only few suggestions. I felt that the study would profit from some contextualization of the narrative, in terms of geographical placements, and more specifically University policies on gender in the region that is the focus of the analyses. The article is written as if it was independent from the context, however, contexts matter because people, and in this case women, can internalize it. Internalization has consequences for both feelings and behavior. Related to this, I would like to see some more elaboration on external validity of these conclusions. Moreover, it would be good that the authors in the first part of the article give some sort of overview of similar work and conclusions of other studies.

As a minor comment, Table 1 title is in Spanish. The rest of the article has no issues of this kind.

Author Response

(The authors gave the same response as above.)

Reviewer 3 Report

Page #

Line #

Comments

1

19

Axiological. Not found in Oxford dictionary online

1

34,41

Should LS not be SL? Be consistent throughout the paper.

1,2

28-72

From the beginning of the paper, the reference names are not included alongside the reference number. From line 73+, the references name are alongside your reference numbers, with some exceptions. It is important to be consistent.

2

47

An open bracket “(“  starts in this line, but I do not find a corresponding closed bracket  ”)”.

2

55,64,88

Should LS not be SL?

Page 1,  shows SL- Sustainable Leadership. You thereafter have used LS? (I guess you imply Leadership for Sustainability ?)

     Spanish and English usage of adjectives are often in a reverse sequence.

     English- use SL.         Be consistent  with your choice. )

2

79

Reference 23, 27,28 have no reference authors’ name shown alongside

2

93

Reference 9  has no reference author’s name shown alongside

3

103

Reference 30  has no reference author’s name shown alongside

3

110-114

Reference 38- 41  have no reference author’s name shown alongside

General 1

Please check the remainder of the paper for the

    “no name alongside reference number “ problem – e.g.   

     References 45 -48 on page 4, lines 153-156; to the end of the paper.

3

142

UA – is this correct, or USA, or UAE, or University of Alicante, or other?

3

148-147

I suggest you insert “leadership “ between the words “women” and “academics”, (or some other suitable word.)

3

148-149

… this suggests… ???

Explain this suggestion, or give a reference for this suggestion.?

4

158-159

“Satisfactions” and “opportunities”  are effectively 2 questions? Are there perhaps 4 questions?

4

171

Insert “available” between “of” and “dates”.

4

185

Is it 3 or 4 questions?

4/5

202-208

Please explain AF and AF%.

5

209

The title line of Table 1 appears to be in Spanish

5

Table 1

The numeric results’ headings are FA and %FA You have reversed the letters FA and AF?)

5

Table 1

You have in effect shown 4 questions in your codes?

5

222

Hyphenate socio-emotional

5

232-233

Delete the last phrase after … you a lot “,they also respect you a lot”.

 It seems to be a repeated phrase. ”

5/6

233-249 et seq.

What re “(Aca_19)” ; “(Aca_26)”; ”(Aca_44)”? etc? Are they author reference numbers, or participants numbers.,  or ?

7

321

“Code” 2  Type of opportunities. Add the word Code

7

329

Add “Sub-code” before the number 2.1

7

332

Should LS not be SL?

8

340

Add “Sub-code” before the number 2.2

8

376

Add “Sub-code” before the number 2.3

9

404

Add “Sub-code” before the number 3.1

9

428-430

This sentence is not clear to me.

10

447

Delete “important” and replace with “useful”

14

     676

Frances (2018) has a reference number 18. Insert this number to your text, as you have been doing before.

14

679

Should LS not be SL?

15

685

LSF was SFL before. Use what you used before.- see page 2,

lines 66 and 67 - ” female sustainable leadership (SFL)

General 2

See lines 9-11 on page 1

Forty-eight leaders participated in the study. A purposive sampling technique was used because the research involved leaders who had held a management position in the university. Were there male and female participants in this group of 48 leaders? Have the male and female responses been compared?

     General 3             References not found in reference list

a.      All the references are in the paper.

b.      There are referencing  issues in the document See notes above, re lines  28-72 of page 1 / 2.

     General 4             References found in Reference list, but not found in text.

All the references are in the text

Author Response

(The authors gave the same response as above.)

Reviewer 4 Report

Authors need to check the English throughout as very difficult to read. Many mistakes

What does LS means? I guess is Sustainable Leadership.

This is an interesting study. However, authors need to identify the literature gap. The sample comes from a single university. How comes to generalize the result. Authors need to find other researchers all over the world to get more respondents on other countries. Now, I am worried about validity of the result.

It is a little pity that no theory back up on your research objectives. Only descriptive results reported. No detail is provided on coding procedure. Now, only simple coding was employed.

Author Response

(The authors gave the same response as above.)

Round 2

Reviewer 1 Report

I commend the authors on a solid revision. Describing research on women in academic leadership as in its infancy seems implausible. There is a good deal of research on this topic, particularly as funded by the National Science Foundation’s ADVANCE initiative. The author should revise that statement and be sure they are citing ADVANCE research. 

Author Response

Dear Reviewer,

Thank you very much for your recommendations, which have undoubtedly improved our work.

Please find attached a document with the resolution to the suggested revisions.

Thank you very much for your

Reviewer 4 Report

Dear authors,

Some improvements. You added more studies but what is the literature gap? You need to talk about this by the end of Introduction section. Then, objective of your study.

Authors only add a theory. What we expect is you have a theoretical framework as a underpinning thing. Some information on coding added but is too little. Sorry, I cannot verify the process.

Author Response

(The authors gave the same response as above.)

Round 3

Reviewer 4 Report

I am fine to the revision